# Relevance of CSF, Serum and Neuroimaging Markers in CNS and PNS Manifestation in COVID-19: A Systematic Review of Case Report and Case Series

**DOI:** 10.3390/brainsci11101354

**Published:** 2021-10-14

**Authors:** Sanjiti Podury, Samiksha Srivastava, Erum Khan, Mihir Kakara, Medha Tandon, Ashish K. Shrestha, Kerri Freeland, Sijin Wen, Shitiz Sriwastava

**Affiliations:** 1Army College of Medical Sciences, New Delhi 110010, India; sanjiti.1997@gmail.com; 2Department of Neurology, Wayne State University, Detroit, MI 48201, USA; samiksha_sami@hotmail.com; 3B.J. Medical College and Civil Hospital, Ahmedabad 380016, India; erum2006@gmail.com; 4Department of Neurology, University of Pennsylvania, Philadelphia, PA 19104, USA; mihir.kakara@pennmedicine.upenn.edu; 5Department of Neurology, University of Pittsburgh Medical Center, Pittsburgh, PA 15213, USA; medhatandon22@gmail.com; 6Kathmandu Medical College, Kathmandu 44600, Nepal; sh_ashish@hotmail.com; 7Department of Biostatistics, West Virginia University, Morgantown, WV 26506, USA; keswails@mix.wvu.edu (K.F.); siwen@hsc.wvu.edu (S.W.); 8Department of Neurology, Rockefeller Neuroscience Institute, West Virginia University, Morgantown, WV 26506, USA; 9West Virginia Clinical and Translational Science Institute, Morgantown, WV 26506, USA

**Keywords:** COVID-19, SARS-CoV-2, ADEM, AHNE, myelitis, GBS, CSF in COVID-19, MRI in COVID-19

## Abstract

Background: The data on neurological manifestations in COVID-19 patients has been rapidly increasing throughout the pandemic. However, data on CNS and PNS inflammatory disorders in COVID-19 with respect to CSF, serum and neuroimaging markers is still lacking. Methods: We screened all articles resulting from a search of PubMed, Google Scholar and Scopus, using the keywords “SARS-CoV-2 and neurological complication”, “SARS-CoV-2 and CNS Complication” and “SARS-CoV-2 and PNS Complication” looking for transverse myelitis, vasculitis, acute disseminated encephalomyelitis, acute hemorrhagic necrotizing encephalitis (AHNE), cytotoxic lesion of the corpus callosum (CLOCC) and Guillain-Barré syndrome (GBS), published between 1 December 2019 to 15 July 2021. Results: Of the included 106 CNS manifestations in our study, CNS inflammatory disorders included transverse myelitis (17, 14.7%), AHNE (12, 10.4%), ADEM (11, 9.5%), CLOCC/MERS (10, 8.6%) and vasculitis (4, 3.4%). Others were nonspecific encephalopathy, encephalitis, seizures and stroke. Most patients were >50 years old (75, 70.8%) and male (64, 65.3%). Most (59, 63.4%) were severe cases of COVID-19 and 18 (18%) patients died. Of the included 94 PNS manifestations in our study, GBS (89, 92.7%) was the most common. Most of these patients were >50 years old (73, 77.7%) and male (59, 64.1%). Most (62, 67.4%) were non-severe cases of COVID-19, and ten patients died. Conclusion: Our comprehensive review of the clinical and paraclinical findings in CNS and PNS manifestations of COVID-19 provide insights on the pathophysiology of SARS-CoV-2 and its neurotropism. The higher frequency and severity of CNS manifestations should be noted by physicians for increased vigilance in particular COVID-19 cases.

## 1. Introduction

As of 28 July 2021, the World Health Organization (WHO) has reported 195 million confirmed cases and 4 million deaths due to COVID-19 [1]. Though it predominantly presents with symptoms and complications of the respiratory system, several cases with neurological manifestations have been reported. “Long COVID” is the term used to describe potential long-term sequelae of COVID-19. In this report, we analyze various thus far reported central nervous system (CNS) and peripheral nervous system (PNS) manifestations of COVID-19, to provide a concise resource for neurologists.

The neurological manifestations of COVID-19 can be studied in two categories:CNS-related manifestations such as acute cerebrovascular disease (ACvD), including ischemic stroke, dural sinus venous thrombosis and hemorrhages, seizures, meningoencephalitis/encephalitis, acute disseminated encephalopathies (ADEM), acute hemorrhagic necrotizing encephalopathy (AHNE), transverse myelitis and cytotoxic lesion of corpus callosum/mild encephalopathy with reversible splenium lesion CLOCC/MERS [2,3,4,5,6,7,8,9,10,11,12,13].Peripheral nervous system (PNS)-related manifestations such as Guillain-Barré syndrome (GBS) and its *Var.* (including acute inflammatory demyelinating polyneuropathy (AIDP), acute motor-sensory axonal neuropathy (AMSAN), acute motor axonal neuropathy (AMAN), Miller Fisher syndrome (MFS), bifacial diplegia (BFP) and other cranial nerve deficits, including polyneuritis cranialis and myasthenia gravis (MG) [14,15,16,17,18,19,20].

Other reported clinical presentations include loss of taste and smell [21,22] and rare manifestations such as nerve pain and skeletal muscle injury such as myopathy/myositis [23,24,25].

Currently, the understanding of the mechanism of these symptoms in COVID-19 patients is lacking, and it is not clear if these symptoms are due to direct viral invasion or an indirect neuroinflammatory response. Intense systemic inflammatory response can lead to disruption of the blood–brain barrier (BBB), causing increased permeability to inflammatory cytokines [26].

It has also been hypothesized that the etiology of GBS is due to the presence of viral spike (S) protein in the cell surfaces of SARS-CoV-2, which binds with angiotensin-converting enzyme 2 receptors and gangliosides containing sialic acid residues such as Ga1NAc residue of GM1. In addition, molecular mimicry due to the cross-reactivity between the viral protein-associated gangliosides and peripheral nerve gangliosides have been proposed [27,28].

## 2. Methods

### 2.1. Study Design

We conducted a thorough literature review in October 2020 using the terms “SARS-CoV-2 and neurological complication”, “SARS-CoV-2 and CNS Complication” and “SARS-CoV-2 and PNS complications”. We searched the PubMed, Google Scholar and Scopus databases for identifying case series and case reports published between 1 December 2019 to 15 July 2021; review articles and consensus statements were excluded from the analysis. We used the preferred reporting items for systematic reviews and meta-analyses (PRISMA) for the display of inclusions and exclusions (Figure 1) [29]. The review protocol was registered with PROSPERO with the registration number **CRD42021268791**. Based on our search criteria, we found the following numbers of articles from PubMed (n = 354), Google Scholar (n = 1780) and Scopus (n = 355). Amongst all, 800 were identified as duplicates. Finally, we screened 1047 articles for title and abstracts, and reviewed full-text literatures in accordance with our study objective after removing 798 articles which were either missing clinical information or did not meet our study objective. We included 140 articles for review for quantitative analysis (Figure 1).

### 2.2. Inclusion Criteria

The inclusion criteria for the published studies included: (1) Patient age ≥ 18 years; (2) COVID-19 diagnosis confirmed by RT–PCR nasopharyngeal or serum antibody test; (3) CSF study findings in COVID-19 and MRI and or CT imaging performed.

### 2.3. Exclusion Criteria

The exclusion criteria for the studies include: (1) Patient age < 18 years; (2) Duplicate articles which involved repetition of cases; (3) Articles in languages other than English; (4) Studies that had no available individual patient data; (5) Editorials.

### 2.4. Quality Assessment

The critical appraisal checklist for case reports provided by the Joanna Briggs Institute (JBI) was used to perform an assessment of the overall quality of case series and case reports.

### 2.5. Data Extraction

Two reviewers independently performed the literature search and missing data were sought by discussion. The reviewers screened titles, abstracts and keywords to check for the inclusion and exclusion criteria. We used the preferred reporting items for systematic review and meta-analysis (PRISMA) for the study. We applied the PRISMA checklist for the article search for case report and case series. The data extracted were included in an Excel spreadsheet; R software was used for analysis, and result were further tabulated.

### 2.6. Data Acquisition

From the selected articles, we extracted the following data for our analysis: study type, date of publication, age, sex, clinical presentation of COVID-19, diagnostic tests for SARS-CoV-2 infection including RT–PCR nasopharyngeal and serum antibodies, CSF markers including cell count, lymphocytes percentage, protein, IL-6, severity of COVID-19 (based on IDSA/ATS criteria) [30] and neuroimaging findings.

### 2.7. Statistical Analysis Methods

We performed demographic analysis on individual patient data including age, sex, severity of COVID-19 cases and fatality of the cases. The pooled descriptive analyses were conducted to assess differences in these markers among groups including severe vs. non-severe COVID-19, fatal vs. non-fatal COVID-19 and CNS vs. peripheral nervous system (PNS) manifestations. Statistical analysis was performed with patient-based data since the patient characteristics and the outcome variable were available for individual patients. In particular, Pearson correlation was used to estimate the correlation between continuous variables. The Fisher exact test and the Wilcoxon rank-sum test were used in the univariate data analysis for categorical and continuous variables, respectively, while the logistic model was used in the multivariate data analysis. All statistical tests were two-sided and a *p*-value < 0.05 implies the statistical significance in this study. Statistical analysis was performed using SAS (version 9.2) and R software (version 3.6.3, R foundation, Vienna, Austria).

## 3. Results

### 3.1. CNS Manifestations

#### 3.1.1. Demographics

Of the included 106 CNS manifestations in our study, common CNS inflammatory disorders were transverse myelitis (n = 17, 14.7%), AHNE (n = 12, 10%), ADEM (n = 11, 9.5%) and CLOCC/MERS (n = 10, 8.6%). Most patients were >50 years old (n = 75, 70.8%) and male (n = 64, 65.3%). Most (n = 59, 63.4%) were severe cases of COVID-19, and (n = 18, 18%) patients had a fatal outcome (Table 1). In COVID-19 patients aged 50 or older, the most common CNS complications were non-specific encephalopathy (n = 18, 24%), seizures (n = 12, 16%), AHNE (n = 9, 12%) and transverse myelitis (n = 9, 12%). Other common manifestations included encephalitis (n = 7, 9%), CLOCC (n = 7, 9%), ADEM (n = 6, 8%) and strokes (n = 5, 7%) (Table 2 and Table 3).

The demographics and frequency of the CNS manifestations are shown in Table 1 and Appendix A. On sex-based analysis of CNS complications in COVID-19 patients, we found that male patients most commonly presented with CLOCC/MERS (n = 9, 90%) and encephalitis (n = 11, 79%). Other common manifestations for males included seizures (n = 9, 69%) and ADEM (n = 7, 64%), while female patients with COVID-19 most commonly presented with stroke (n = 2, 50%) and vasculopathies (n = 2, 50%). Other manifestations for female patients included non-specific encephalopathy (n = 9, 47%) and transverse myelitis (n = 8, 47%) (please refer Table 2 and Table 3). Figure 2 shows a pie chart of various CNS-manifestation post COVID-19 infection.

In our study, the average latency period between the positivity of COVID-19 to symptoms onset of CNS disorder was 12.8 days, which was in correspondence with other reported study in which CNS involvement occurred over 1 week after the onset of COVID-19 symptoms [31]. The latency period ranged in this review for CNS symptoms occurring within the same day to up until 8 weeks after the onset of COVID-19 symptoms.

CNS, central nervous system; ADEM, acute disseminated encephalomyelitis; AHNE, acute hemorrhagic necrotizing encephalitis; CLOCC, cytotoxic lesion of the corpus callosum; MERS, mild encephalitis/encephalopathy with reversible splenial lesion.

#### 3.1.2. CSF Analysis

CSF RT–PCR were reported in 78 patients, among which five cases had positive results, which included cases of AHNE, meningoencephalitis, acute cerebellitis, CNS demyelination and ADEM [32,33,34,35,36]. CSF IgM and IgG antibodies were reported in 28 cases, in which IgG was present in cases of AHNE and MERS [13,32], while IgM was reported to be positive in three cases of encephalopathy [37].

A CSF protein level greater than 45 mg/dL was considered elevated. This was noted in patients with AHNE (100%), transverse myelitis (75%), encephalitis (71%) and ADEM (70%). The CSF protein in patients with AHNE showed a significant elevation (*p*-value of 0.01) in all of the nine patients who underwent CSF analysis. A cell count of greater than 5 cells/mm^3^ is considered to be elevated; this was most commonly seen in patients with transverse myelitis (69%) and ADEM (40%). Patients with transverse myelitis showed statistically significant elevation in their CSF cell count (*p*-value of 0.001); that is 9 out of 13 patients (Table 2).

#### 3.1.3. Severity

On evaluating the patients and classifying them based on IDSA/ATS severity criteria [28], we found that patients with AHNE (100%) and encephalitis (83%) had severe COVID-19 infection more commonly. Infection in patients with AHNE was statistically significant (*p*-value of 0.012), as all the patients had severe infection (Table 2 and Table 4).

#### 3.1.4. Fatality

The majority of the fatal outcomes were seen with AHNE (55%) and vasculitis (33%). The fatality outcome in AHNE was statistically significant with a *p*-value of 0.004 (6 out of the 11 patients) (Table 2).

The predominant cause of death in CNS-affected patients was cardiorespiratory failure. However, there were two cases of reported fatality due to absent brainstem reflexes [37,38]. Also, cases reported to have fatality likely due to bilateral uncal herniation and tonsillar herniation [39,40] (Appendix A).

#### 3.1.5. Serum Markers

We evaluated serum markers like d-dimer and C-reactive proteins. These markers have traditionally been used in following pro-inflammatory and hypercoagulable states. A value greater than 500 ng/mL was used as the cutoff for elevated serum d dimer levels. Elevated levels were most commonly seen in patients with AHNE (27%) and transverse myelitis (17%). However, these findings were not statistically significant in any of the CNS complication. CRP levels greater than 8mg/mL are considered elevated, and a statistically significant elevation was noted in patients with transverse myelitis (*p* value 0.046) and vasculitis (*p*- value 0.011) (Table 5). The elevation of these serum markers can be attributed to the SARS-CoV-2 infection. A study by Huang et al. noted the diagnostic value of serum CRP  ≥ 10 mg/L for a composite poor outcome in COVID-19, with 88% specificity. Analyzing for d-dimer, Gao et al. found elevated d-dimer levels to have 82.1% specificity for predicting the severity of COVID-19 infection.

### 3.2. PNS Manifestations

Of the included 94 PNS manifestations in our study, GBS (n = 89, 92.7%) was the most common. Most patients were >50 years old (n = 73, 77.7%) and male (n = 59, 64.1%). Most (n = 62, 67.4%) were non-severe cases of COVID-19, and ten patients died. The demographics and frequency of the PNS manifestations are shown in Table 6 and Appendix A. Figure 3 shows a pie chart of various PNS manifestations post COVID-19 infection.

#### 3.2.1. Demographics

Most patients with GBS (81%) were aged over 50. All patients with polyneuritis cranialis and 50% of patients with facial palsy were aged less than 50. A statistical significance could not be ascribed to polyneuritis cranialis and facial palsy due to the inadequate number of patients in the two subgroups. Among 56 of 88 patients with GBS (64%) were males, and equal numbers of males and females were affected in case of facial paralysis and polyneuritis cranialis. The latency period for PNS disorders (mostly GBS) after COVID-19 infection was 14.9 days. According to Sriwastava S. et al., latency was around two weeks for GBS onset from COVID-19 infection [16]. This study showed the latency period ranging from the same day to 13 weeks after a positive COVID-19 test (Table 6 and Appendix A).

#### 3.2.2. CSF Analysis

CSF RT–PCR were reported for 40 patients and were negative for all cases. CSF IgM and IgG antibodies were also reported in 35 cases, but only a case of AIDP had IgM and IgG positive in CSF, whereas RT–PCR in CSF was negative [41].

A CSF protein level greater than 45 mg/dl is considered elevated. Elevated levels were noted in all patients with polyneuritis cranialis and 82% of patients with GBS. A cell count greater than 5 cells/mm^3^ is considered elevated, and most patients among all three subgroups showed cell counts of less than 5 cells/mm^3^. An elevated CSF cell count was seen in 5 out of 75 patients with GBS (7%) (please refer to Table 7).

#### 3.2.3. Serum Markers

A d-dimer level greater than 500 ng/mL and a CRP value greater than 8 mg/mL is considered elevated. All patients with GBS (100%) showed an elevation in CRP and d-dimer levels; however, the findings were not significant in patients with facial palsy and polyneuritis cranialis (Table 8).

#### 3.2.4. Severity

The patients were categorized according to the IDSA/ATS classification. All patients with facial paralysis and polyneuritis cranialis had non-severe infection; however, 30 out of 87 patients with GBS (34%) were classified as severe COVID-19 infection (Table 6 and Appendix A).

#### 3.2.5. Fatality

A 12% fatality (10 out of 84 cases) was noted among patients with GBS. None of the patients with facial paralysis or polyneuritis cranialis had a fatal outcome (Table 6 and Table 9 and Appendix A).

The most common cause of death was found to be due to ARDS or respiratory failure but there were two cases fatality was reported due to GBS-associated autonomic dysfunction or respiratory failure [42].

## 4. Neuroimaging Findings

### 4.1. CNS Disorders

There were 106 cases of COVID-19 patients who had CNS complications. Few cases reported had more than one CNS diagnosis, leading to 115 CNS disorders [33,37,43,44] (Table 1). Among them, relevant MRI findings reported were for encephalitis (26/106), transverse myelitis/LETM (17/106), non-specific encephalopathy (15/106), ADEM/AHLE (14/106), seizures (11/106), AHNE (8/106), CLOCC/MERS (6/106), PRES (4/106), stroke (4/106), vasculopathy/vasculitis (3/106) and cerebellitis (1/106).

The total number of MRIs of spinal cord performed was 21, in which the predominant findings were intramedullary hyperintensity and had a diagnosis of transverse myelitis; among them, three had normal findings [43,45,46]. There were a total of 42 cases who underwent a CT scan of the head, among which 80% cases had normal/unremarkable CT head results. There were four cases of AHNE which showed abnormal CT findings. Symmetrical hypodensities in thalami, midbrain and right cerebellar hemorrhages were reported [32,47], whereas another two cases had hydrocephalus with hemorrhage in the bilateral parietal and occipital regions, bilateral thalami and the sub-arachnoid spaces, with intraventricular extension [10,48] (Appendix A). There was one study that used PET, as reported by Delorme C et al., a case series consisting of four patients with COVID-19-related encephalopathy with brain FDG-PET/CT patterns indicating frontal hypometabolism and cerebellar hypermetabolism.

### 4.2. PNS Disorders

Out of the 94 total cases studied in PNS disorders, 55 included either brain or spine MRIs or both brain and spine MRIs. This included cases where MRIs of the entire spine (cervical, thoracic and lumbosacral spine) were obtained. In the remaining 39 cases, MRIs were either not reported or not performed.

The most common findings reported on MRIs of the spine were abnormal enhancement of the cauda equina/lumbar spinal nerve root enhancement (n = 6) [15,49,50,51,52] and abnormal hyperintensity on T2/STIR sequences along with enhancement of postganglionic roots supplying the brachial and lumbar plexus (n = 2) [53,54], in reported cases of GBS.

Similarly, the most common findings on brain MRIs were cranial nerve involvement as seen in the following cases. The most common nerves involved were the facial nerve [15,55] and the oculomotor nerve [56,57]; however, multiple cranial nerve involvement has been reported, including CN VII, CNV VI, CN III and III, V, VI, VII and VIII [53,58]. MRI findings included abnormal thickening and enhancement of cranial nerve III, abnormal bilateral enhancement of cranial nerve V and abnormal bilateral enhancement of VI and VII has also been reported [52]. An additional case demonstrating leptomeningeal enhancement involving the brainstem and cervical spine was also reported [59] (Appendix A).

## 5. Discussion

During the course of the COVID-19 pandemic, a number of cases of central and peripheral nervous system involvement in COVID-19 have been reported as associated with the SARS-CoV-2 infection. In this review paper, we evaluated and reviewed all of the 106 cases with CNS and 94 cases with PNS involvement that have been reported in the literature to date. We focused only on acute (defined for the purposes of this review as within four weeks of COVID-19 symptoms) and sub-acute onset of neurological symptoms (defined as 4–8 weeks of onset of COVID-19 symptoms). Firstly, we found that the most common clinical presentations reported were from underlying CNS inflammatory and immune-related processes. In CNS involvement, these were in the form of transverse myelitis, ADEM, CLOCC/MERS and AHNE, whereas in case of PNS involvement, the most common manifestation reported was GBS.

With a global pandemic that has affected 212 million people worldwide, there is always a chance that natural incidence of the reported diagnoses can be coincident with a COVID-19 diagnosis, and it is almost always impossible to conclusively link the viral infection by itself or the ensuing inflammatory and/or autoimmune process to the neurological presentation. But in the reported cases, we did find most authors attempting to delineate this based on the best available evidence. The wide involvement of the nervous system was attributed by most of the authors to be largely from para or post-infectious inflammatory responses due to cytokine release or molecular mimicry. [27,28,60,61,62]. The majority of cases were attributed to post-infectious mechanisms through suggestions of autoimmunity by mechanisms like molecular mimicry. Proving this association is even harder in the absence of advanced immunological techniques like B-cell or T-cell receptor sequencing.

When present, evidence of the virus in the CSF may present the best proof in associating neurological manifestations of the infection. We found that 78 out of 106 (73%) patients with CNS involvement underwent RT–PCR testing for the virus in the CSF, and a positive result was seen in five patients (6%). These five patients had clinical presentations of one each case of AHNE, meningoencephalitis, cerebellitis, CNS demyelination disorder and ADEM in this review study [32,33,34,35,36].

Upon further review of the literature, we found the following cases of positive SARS-CoV-2 CSF RT–PCR: three cases of meningoencephalitis [63,64,65], two cases of seizure [66,67] and cases of headache with impaired consciousness [68]; none of these cases had IgM or IgG reports available in CSF. Technical factors and rapid viral clearance from the CSF can make it challenging to isolate the virus [69] and the test is non-specific due to the possibility of contamination of the sample by airborne virus or blood contamination [70].

Serological responses in CSF can be another marker of presence of SARS-CoV-2 in the CNS compartment. The presence of IgM antibodies in the CSF could be due to intrathecal synthesis or due to the breakdown of the blood–brain barrier caused by high levels of circulating proinflammatory cytokines due to SARS-CoV-2 infection [71]. Animal and human studies have shown that human coronaviruses are capable of neurovirulence and neuroinvasion. SARS-CoV-2 belongs to the same family and shows similar receptor- binding properties, explaining the neurological manifestations [72].

Among the patients reported, SARS-CoV-2 antibodies in CSF were present in 5 out of 28 cases, including three cases in which IgM positivity was seen with all encephalopathy [37], and IgG in two cases with a case of AHNE and MERS [13,32].

On further exploration of the literature in this study that evaluated 106 cases with CNS manifestations of COVID-19, the most common neuroimaging findings were reported for CNS inflammatory disorders (transverse myelitis/LETM (16%), ADEM/AHLE (13.2%), AHNE (7.54%), CLOCC/MERS (5.66%), vasculopathy/vasculitis (2.83%)). All patients with acute hemorrhagic necrotizing encephalopathy (AHNE) had a severe infection, while only 52.68% of patients without AHNE developed severe COVID-19 disease (overall severity was not known in 13 cases) (refer to Table 2). Additionally, we observed that a greater proportion of patients with AHNE reported a fatal COVID-19 condition (54.54%) (out of 12 cases of AHNE, fatality was not known in one case) compared to those with other CNS manifestations (12%) (unknown outcome in 6 CNS cases) (*p* = 0.004) (refer to Table 2). This could be because patients with severe disease developed severe neurovascular injury. Cytokine storm with dysregulation of BBB predisposes to AHNE [73]. AHNE may be secondary to para infectious and hyper immune response to COVID-19. Rapid progression of encephalopathy is accompanied by distinct neuroimaging features of multifocal discussion restriction and micro hemorrhages of cortex, subcortical and atrioventricular white matter, the brain stem and infratentorial regions [74,75]. Hence, AHNE should be meticulously considered in patients with severe COVID-19 infections.

The first case of meningoencephalitis reported by Moriguchi et al. in March 2020 was negative for SARS-CoV-2 on nasopharyngeal analysis but was positive in CSF analysis [33]. An MRI demonstrated signal abnormality in the hippocampus with associated ventriculitis. A re-emergence of patients presenting with meningoencephalitis has been observed since September 2020. Many of these cases demonstrated a viral infectious picture in the CSF [76,77,78]. Though the mechanism of meningoencephalitis is not completely understood in COVID-19, MRI changes were similar to other viral causes of meningoencephalitis in the literature [79].

Acute transverse myelitis presents with varied signs and symptoms of sensory, motor and autonomic dysfunction. Among the patients who underwent a CSF study, cell counts were significantly elevated in 69% of patients with TM, while the remaining CSF parameters were not significant. Patients presented with symptoms of sensory, motor and autonomic dysfunction [80,81,82,83,84,85,86]. Pathological changes in the spinal cord tract were seen on imaging, as a result of focal inflammation. As this disorder can lead to debilitating effects and permanent disability, it is important to recognize it early and distinguish it from other neurological entities. The diagnosis of transverse myelitis involves characteristic clinical presentation of bilateral signs and symptoms with a clearly defined sensory level, in addition to the evidence on neuroimaging, CSF and serologic studies as per Proposed Diagnostic Criteria and Nosology of Acute Transverse Myelitis, 2002 [87].

Cerebrovascular accidents (ischemic and hemorrhagic) have been a commonly reported neuroradiologic abnormality among COVID-19 patients. Elderly populations with comorbidities have a predilection for this complication, as reported in a prior study [88]. The prevalence of CVA in elderly patients with COVID-19 infection was reported to be 5.7% [4,88,89,90]. The majority of the strokes reported were ischemic, as noted in a study by Mao [4,90,91,92]. In this review, five cases of infarct were reported post COVID-19 infection. Four of the cases were arterial infarcts, ranging from small lacunar infarcts to large territory infarcts [38,52]. One of these demonstrated occlusive thrombus in the right internal carotid artery resulting in multiple acute infarcts [37].

Patients with CLOCC associated with COVID-19 presented with atypical neurological manifestations of COVID 19 such as agitation, disorientation, violent behavior and sometimes even hallucinations [93,94,95,96]. The diagnosis of CLOCC is based on MR imaging demonstrated diffusion restriction and non-enhancing lesions mainly in the splenium of the corpus callosum.

ADEM cases were also reported post COVID-19 or as an atypical presentation of COVID-19; however, these did not always have the hallmark perivenular, intracortical and cortical subpial demyelination [6,7]. There were four cases of vasculitis reported with abnormal neuroimaging findings. MRI showed multiple hyperintense lesions involving the deep and subcortical white matter on both hemispheres, as well as the thalami, basal ganglia and brainstem [97,98]. A case by Vaschetto R. et al. reported restriction diffusion in a parietal and parieto-occipital region and at the pons.

In our study, the CNS manifestations were more frequent and severe than the PNS manifestations. This can be attributed to increased brain vulnerability to hypoxia compared to peripheral nerves, or higher expression of ACE2 receptors in neuronal soma than axons and dendrites. Although exact pathophysiology has not been delineated yet, it is suspected that a combined direct and indirect mechanism play a role in developing CNS and PNS involvement [99,100].

Out of 96 patients who presented with involvement of the PNS 40 underwent evaluation for CSF RT–PCR and one tested positive for SARS-CoV-2, as reported by Masuccio et al. Elevated CSF cell count levels were noted in 4 out of 84 patients who underwent CSF cell count analysis, and elevated protein levels were seen in 64 out of 84 patients who underwent CSF protein analysis. On evaluation of CSF SARS-CoV-2 immunoglobulins, one case was positive for IgG and as well as positive for IgM immunoglobulins [41]. The most common neuroimaging findings in PNS manifestation were lumbar spinal nerve root enhancement and abnormal hyperintensity with enhancement of the post-ganglionic brachial and lumbar nerve root (13/30, 43.33%) (refer to Appendix A). GBS was the most common PNS manifestation of COVID-19, presenting clinically with paresthesia and lower limb weakness. Among GBS, 30 cases were severe (32.6%) (unknown severity in two cases) and 10 cases were fatal (10/90, 11.11%) (unknown outcome in four cases) (refer to Table 9 and Appendix A).

The first presentation of COVID-19-associated GBS was reported by Zhao et al. in a 61-year-old female who presented as GBS and was later tested positive for COVID-19 [101]. Toscano G. et al. reported four patients who were positive for COVID-19 and later developed neurological manifestations of GBS, whereas Virani et al. reported patients who were diagnosed with COVID-19 after developing neurological manifestations [14,15]. Cases reported by Mozhdehipanah et al., and Ebrahimzadeh et al., developed GBS two to four weeks after testing positive for COVID-19 [102,103].

Reports of Miller Fisher syndrome, a variant of GBS, in COVID-19 patients have been published with additional symptoms like truncal ataxia and dysplasia apart from the classical triad of ataxia, areflexia and ophthalmoplegia [17,56,57,104,105,106,107,108]. Among all the PNS disorders in COVID-19 infection, there were 11 cases which reported bilateral facial nerve palsy [20,50,102,103,105,109,110,111,112,113,114,115]. Guilmot et al. and Homma et al. reported cases with unilateral facial nerve involvement [52,116]. The MRI findings in two cases showed abnormal enhancement in cranial nerve III, V, VI and VII [52,58]. The most frequent GBS variant in association with COVID-19 in this review was AIDP (46/90, 51%) which is consistent with the other literature available [98,99,117]; the remainder were AMSAN (17/90, 18.8%), MFS (10/90, 11.1%), AMAN (5/90, 5.5%) and BFP (4/90, 4.4%). In seven cases of GBS, the variant was not specified (refer to Appendix A).

Our study had several limitations. First, a subset of neuroimaging findings, particularly those in critically ill elderly patients, may be due to comorbidities or other factors and may not be directly related to the COVID-19 infection. Second, underreporting could be the cause of biased findings, especially those for milder disease, where neuroimaging is not generally preferred. Third, as this is still an emerging disease, and there are only a few hundred cases in the literature on neurological manifestations in COVID-19 (refer to Appendix A), we believe further studies are still required to prove the validity of our results, as bias can be a contributing factor. Fourth, generalization from our results should be done carefully based on our study design limitations. International collaborative efforts with registries made specifically for neurology might increase our knowledge of CNS and PNS findings of COVID-19. Finally, to determine the frequency of imaging must also be challenging; hence, these findings should be interpreted with caution.

## 6. Conclusions

There has been widespread reporting of neurological manifestations following COVID-19 infection due to the involvement of the nervous system directly or indirectly. After a categorical analysis of CNS and PNS involvement, we were able to extract a complete evaluation of CSF, serum and neuroimaging relevance of neurological impairment in SARS-CoV-2-infected patients. In our systematic review, we have demonstrated the clinical and paraclinical findings of the central and peripheral nervous system, after a comprehensive overview of scientific literature, to provide a concise resource for neurologists to review. Further studies exploring this at multiple centers will consolidate our knowledge and help formulate diagnostic and treatment guidelines for neurological manifestations of COVID-19.

## Figures and Tables

**Figure 1 brainsci-11-01354-f001:**
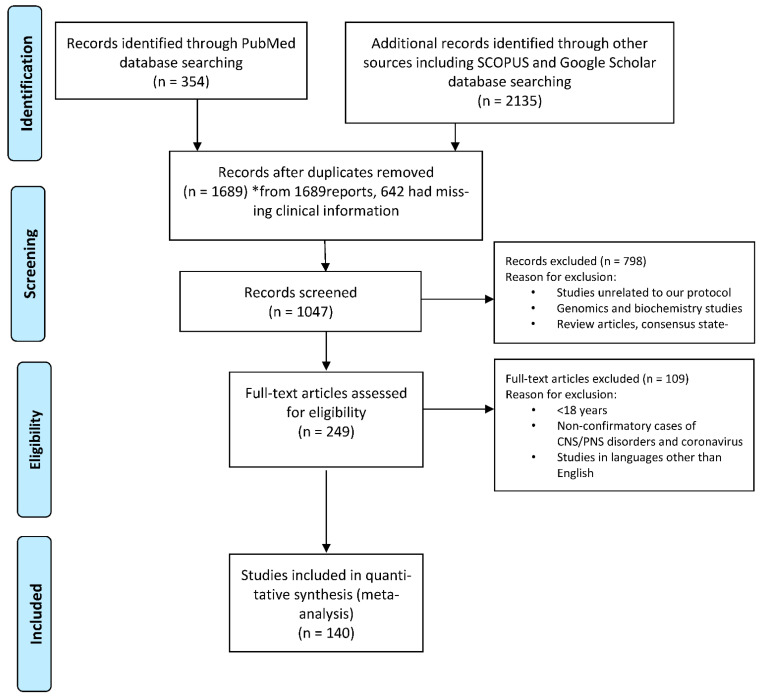
Preferred reporting items for systematic reviews and meta-analyses (PRISMA) flow diagram. * 642 missing data.

**Figure 2 brainsci-11-01354-f002:**
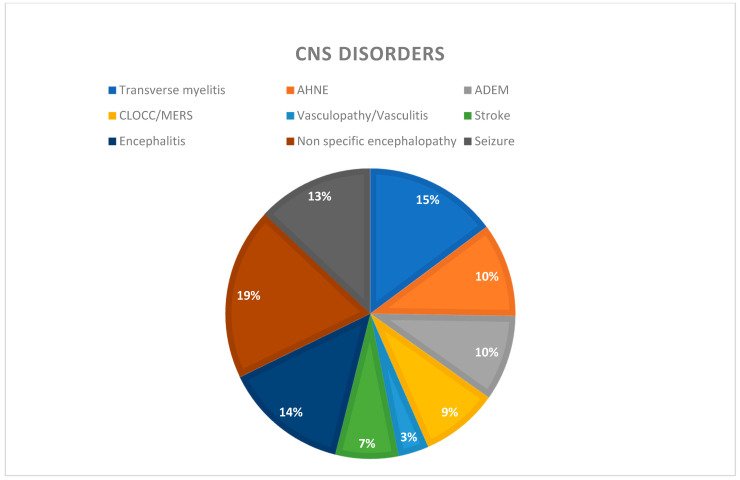
Pie chart showing distribution of various CNS disorders post COVID-19.

**Figure 3 brainsci-11-01354-f003:**
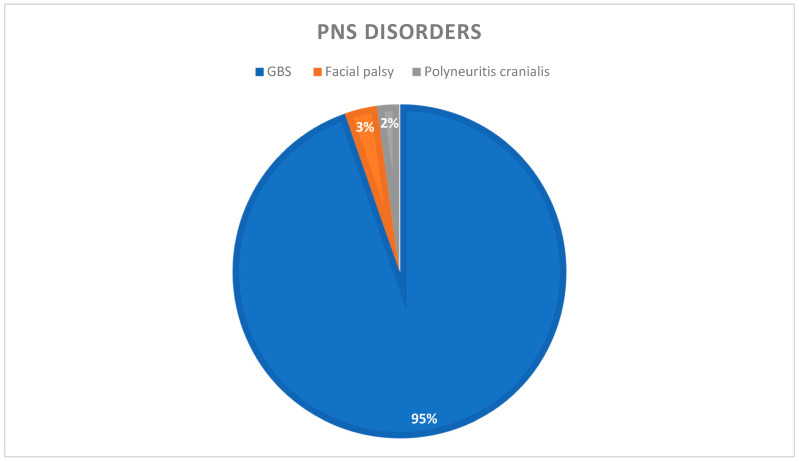
Pie chart showing distribution of various PNS disorders post COVID-19. GBS, Guillain–Barré syndrome.

**Table 1 brainsci-11-01354-t001:** Demographics and CNS manifestations in the 106 patients.

Variables	N (%)
**Age**	
>50	75 (70.8)
≤50	31 (29.2)
* **Sex**	
Male	64 (65.3)
Female	34 (34.7)
**● CNS manifestation**	
Transverse myelitis	17 (14.7)
AHNE	12 (10.4)
ADEM	11 (9.5)
CLOCC/MERS	10 (8.6)
Vasculopathy/ Vasculitis	4 (3.4)
Stroke	8 (7)
Encephalitis	16 (13.8)
Non-specific Encephalopathy	22 (19.1)
Seizure	15 (12.9)
** **COVID-19 Severity**	
Non-severe	34 (36.6)
Severe	59 (63.4)
♦ **Outcome**	
Non-fatal	82 (82)
Fatal	18 (18)

* Gender unidentified in 6 cases, ** Severity not available in 13 cases, ♦ outcomes were not available in 6 cases; ● CNS manifestation some cases had more than one diagnosis.

**Table 2 brainsci-11-01354-t002:** Comparison of various CSF markers and demographics, severity and fatality by CNS manifestation.

	Stroke				Encephalitis				Non-Specific Encephalopathy				Seizures				Vasculopathy/Vasculitis			
* Variables	0	1	Total	Fisher Test	0	1	Total	Fisher Test	0	1	Total	Fisher Test	0	1	Total	Fisher Test	0	1	Total	Fisher Test
	n (%)	n (%)	n (%)	*p*-Value	n (%)	n (%)	n (%)	*p*-Value	n (%)	n (%)	n (%)	*p*-Value	n (%)	n (%)	n (%)	*p*-Value	n (%)	n (%)	n (%)	*p*-Value
CSF protein																				
>45	54(57)	5(62)	59(57.3)	1	50(56)	10(71)	60(57.7)	0.385	52(63)	7(33)	59(57.3)	0.024	53(60)	7(47)	60(57.7)	0.404	58(57)	2(67)	60(57.7)	1
≤45	41(43)	3(38)	44(42.7)		40(44)	4(29)	44(42.3)		30(37)	14(67)	44(42.7)		36(40)	8(53)	44(42.3)		43(43)	1(33)	44(42.3)	
CSF cell count																				
>5	25(28)	2(25)	27(27.8)	1	25(30)	2(14)	27(27.6)	0.338	23(29)	3(17)	26(26.8)	0.383	24(28)	3(25)	27(27.6)	1	26(28)	1(25)	27(27.6)	1
≤5	64(72)	6(75)	70(72.2)		59(70)	12(86)	71(72.4)		56(71)	15(83)	71(73.2)		62(72)	9(75)	71(72.4)		68(72)	3(75)	71(72.4)	
CSF Lymphocyte Percent																				
>50	28(61)	1(33)	29(59.2)	0.559	26(68)	3(27)	29(59.2)	0.033	26(59)	2(50)	28(58.3)	1	27(60)	2(50)	29(59.2)	1	28(60)	1(50)	29(59.2)	1
≤50	18(39)	2(67)	20(40.8)		12(32)	8(73)	20(40.8)		18(41)	2(50)	20(41.7)		18(40)	2(50)	20(40.8)		19(40)	1(50)	20(40.8)	
Severity																				
Non-Severe	32(37)	2(33)	34(37)	1	32(40)	2(17)	34(36.6)	0.199	28(37)	6(33)	34(36.6)	1	32(38)	2(22)	34(36.6)	0.478	33(37)	1(33)	34(36.6)	1
Severe	54(63)	4(67)	58(63)		49(60)	10(83)	59(63.4)		47(63)	12(67)	59(63.4)		52(62)	7(78)	59(63.4)		57(63)	2(67)	59(63.4)	
Fatality																				
Non-Fatal	77(82)	4(80)	81(81.8)	1	73(83)	9(75)	82(82)	0.448	65(82)	17(85)	82(82.8)	1	68(80)	14(93)	82(82)	0.295	80(82)	2(67)	82(82)	0.452
Fatal	17(18)	1(20)	18(18.2)		15(17)	3(25)	18(18)		14(18)	3(15)	17(17.2)		17(20)	1(7)	18(18)		17(18)	1(33)	18(18)	
Age																				
≥50	69(70)	5(71)	74(70.5)	1	68(72)	7(58)	75(70.8)	0.327	57(69)	18(82)	75(71.4)	0.293	63(69)	12(80)	75(70.8)	0.545	74(73)	1(25)	75(70.8)	0.074
<50	29(30)	2(29)	31(29.5)		26(28)	5(42)	31(29.2)		26(31)	4(18)	30(28.6)		28(31)	3(20)	31(29.2)		28(27)	3(75)	31(29.2)	
Sex																				
Male	62(66)	2(50)	64(65.3)	0.608	53(63)	11(79)	64(65.3)	0.367	54(69)	10(53)	64(66)	0.187	55(65)	9(69)	64(65.3)	1	62(66)	2(50)	64(65.3)	0.608
Female	32(34)	2(50)	34(34.7)		31(37)	3(21)	34(34.7)		24(31)	9(47)	33(34)		30(35)	4(31)	34(34.7)		32(34)	2(50)	34(34.7)	
	Transverse Myelitis					AHNE					ADEM			CLOCC/MERS				** **Others**		
CSF_ Protein																				
>45	48(55)	12(75)	60(57.7)	0.172	51(54)	9(100)	60(58.3)	0.01	53(56)	7(70)	60(57.7)	0.512	57(60)	3(33)	60(57.7)	0.163	46(58)	13(57)	59(57.8)	1
≤45	40(45)	4(25)	44(42.3)		43(46)	0(0)	43(41.7)		41(44)	3(30)	44(42.3)		38(40)	6(67)	44(42.3)		33(42)	10(43)	43(42.2)	
CSF_ Cell Count																				
>5	18(21)	9(69)	27(27.6)	0.001	23(26)	3(33)	26(26.8)	0.698	23(26)	4(40)	27(27.6)	0.456	27(30)	0(0)	27(27.6)	0.185	24(32)	3(14)	27(28.1)	0.108
≤5	67(79)	4(31)	71(72.4)		65(74)	6(67)	71(73.2)		65(74)	6(60)	71(72.4)		64(70)	7(100)	71(72.4)		50(68)	19(86)	69(71.9)	
CSF_lymphocyte Percent																				
>50	19(51)	10(83)	29(59.2)	0.089	25(58)	4(67)	29(59.2)	1	25(60)	4(57)	29(59.2)	1	29(62)	0(0)	29(59.2)	0.162	22(55)	7(78)	29(59.2)	0.277
≤50	18(49)	2(17)	20(40.8)		18(42)	2(33)	20(40.8)		17(40)	3(43)	20(40.8)		18(38)	2(100)	20(40.8)		18(45)	2(22)	20(40.8)	
♦ Severity																				
Non-Severe	21(28)	13(76)	34(36.6)	0	33(40)	0(0)	33(35.9)	0.012	30(37)	4(36)	34(36.6)	1	31(37)	3(33)	34(36.6)	1	27(38)	7(37)	34(37.4)	1
Severe	55(72)	4(24)	59(63.4)		49(60)	10(100)	59(64.1)		52(63)	7(64)	59(63.4)		53(63)	6(67)	59(63.4)		45(62)	12(63)	57(62.6)	
Fatality																				
Non-Fatal	68(82)	14(82)	82(82)	1	76(86)	5(45)	81(81.8)	0.004	73(81)	9(90)	82(82)	0.685	74(82)	8(80)	82(82)	1	65(84)	17(81)	82(83.7)	0.742
Fatal	15(18)	3(18)	18(18)		12(14)	6(55)	18(18.2)		17(19)	1(10)	18(18)		16(18)	2(20)	18(18)		12(16)	4(19)	16(16.3)	
Age																				
≥50	66(74)	9(53)	75(70.8)	0.089	65(70)	9(75)	74(70.5)	1	69(73)	6(55)	75(70.8)	0.292	68(71)	7(70)	75(70.8)	1	58(68)	16(80)	74(70.5)	0.416
<50	23(26)	8(47)	31(29.2)		28(30)	3(25)	31(29.5)		26(27)	5(45)	31(29.2)		28(29)	3(30)	31(29.2)		27(32)	4(20)	31(29.5)	
Sex																				
Male	55(68)	9(53)	64(65.3)	0.27	56(66)	7(58)	63(64.9)	0.748	57(66)	7(64)	64(65.3)	1	55(62)	9(90)	64(65.3)	0.158	53(65)	9(60)	62(64.6)	0.771
Female	26(32)	8(47)	34(34.7)		29(34)	5(42)	34(35.1)		30(34)	4(36)	34(34.7)		33(38)	1(10)	34(34.7)		28(35)	6(40)	34(35.4)	

Fisher exact test and Wilcoxon rank-sum test were used in the univariate data analysis for categorical and continuous variables, respectively, while logistic model was used in the multivariate data analysis. All statistical tests were two-sided and a *p*-value < 0.05 implies the statistical significance in this study. CSF, cerebrospinal fluid; ADEM, acute disseminated encephalomyelitis; AHNE, acute hemorrhagic necrotizing encephalitis; CLOCC, cytotoxic lesion of the corpus callosum; MERS, mild encephalitis/encephalopathy with reversible splenial lesion. * Variables take 1 if yes, take 0 if no. ** Other = PRES, posterior reversible encephalopathy syndrome; MOGAD, myelin oligodendrocytes glycoprotein antibody disease; delirium. ♦ Severity based on Infectious Diseases Society of America/American Thoracic Society guidelines.

**Table 3 brainsci-11-01354-t003:** Comparison of CNS manifestation by age.

	Age			
Variables	50+	<50	Total	Fisher Test
	n (%)	n (%)	n (%)	*p*-Value
Stroke				
0	69(93)	29(94)	98(93.3)	1
1	5(7)	2(6)	7(6.7)	
Encephalitis				
0	68(91)	26(84)	94(88.7)	0.327
1	7(9)	5(16)	12(11.3)	
Encephalopathy				
0	57(76)	26(87)	83(79)	0.293
1	18(24)	4(13)	22(21)	
Seizures				
0	63(84)	28(90)	91(85.8)	0.545
1	12(16)	3(10)	15(14.2)	
CNS demyelination				
0	74(99)	28(90)	102(96.2)	0.074
1	1(1)	3(10)	4(3.8)	
Transverse Myelitis				
0	66(88)	23(74)	89(84)	0.089
1	9(12)	8(26)	17(16)	
AHNE				
0	65(88)	28(90)	93(88.6)	1
1	9(12)	3(10)	12(11.4)	
ADEM				
0	69(92)	26(84)	95(89.6)	0.292
1	6(8)	5(16)	11(10.4)	
CLOCC/MERS				
0	68(91)	28(90)	96(90.6)	1
1	7(9)	3(10)	10(9.4)	

CSF, cerebrospinal fluid; ADEM, acute disseminated encephalomyelitis; AHNE, acute hemorrhagic necrotizing encephalitis; CLOCC, cytotoxic lesion of the corpus callosum; MERS, mild encephalitis/encephalopathy with reversible splenial lesion.

**Table 4 brainsci-11-01354-t004:** Comparison of various demographics, severity, fatality and CNS manifestation by CSF markers.

	CSF Protein				Elevated Cell Count				Lymphocytes			
	High	Low			Yes	No						
* Variables	>45	≤45	Total	Fisher Test	>5	≤5	Total	Fisher Test	>50%	≤50%	Total	Fisher Test
	n (%)	n (%)	n (%)	*p*-Value	n (%)	n (%)	n (%)	*p*-Value	n (%)	n (%)	n (%)	*p*-Value
Age												
50+	41(75)	29(74)	70(74.5)	1.000	13(54)	51(80)	64(72.7)	**0.030**	11(48)	12(63)	23(54.8)	0.366
<50	14(25)	10(26)	24(25.5)		11(46)	13(20)	24(27.3)		12(52)	7(37)	19(45.2)	
Sex												
Male	35(67)	20(59)	55(64)	0.494	14(58)	37(66)	51(63.8)	0.613	11(50)	13(65)	24(57.1)	0.366
Female	17(33)	14(41)	31(36)		10(42)	19(34)	29(36.2)		11(50)	7(35)	18(42.9)	
♦ Severity												
Non-severe	17(33)	13(41)	30(36.1)	0.639	15(68)	15(27)	30(39)	**0.002**	15(68)	4(22)	19(47.5)	**0.005**
Severe	34(67)	19(59)	53(63.9)		7(32)	40(73)	47(61)		7(32)	14(78)	21(52.5)	
Fatality												
Non-fatal	41(76)	33(87)	74(80.4)	0.286	18(72)	52(85)	70(81.4)	0.221	16(73)	17(85)	33(78.6)	0.460
Fatal	13(24)	5(13)	18(19.6)		7(28)	9(15)	16(18.6)		6(27)	3(15)	9(21.4)	
Stroke												
0	54(92)	41(93)	95(92.2)	1.000	25(93)	64(91)	89(91.8)	1	28(97)	18(90)	46(93.9)	0.559
1	5(8)	3(7)	8(7.8)		2(7)	6(9)	8(8.2)		1(3)	2(10)	3(6.1)	
Encephalitis												
0	50(83)	40(91)	90(86.5)	0.385	25(93)	59(83)	84(85.7)	0.338	26(90)	12(60)	38(77.6)	0.033
1	10(17)	4(9)	14(13.5)		2(7)	12(17)	14(14.3)		3(10)	8(40)	11(22.4)	
Encephalopathy												
0	52(88)	30(68)	82(79.6)	**0.024**	23(88)	56(79)	79(81.4)	0.383	26(93)	18(90)	44(91.7)	1
1	7(12)	14(32)	21(20.4)		3(12)	15(21)	18(18.6)		2(7)	2(10)	4(8.3)	
Seizures												
0	53(88)	36(82)	89(85.6)	0.404	24(89)	62(87)	86(87.8)	1	27(93)	18(90)	45(91.8)	1
1	7(12)	8(18)	15(14.4)		3(11)	9(13)	12(12.2)		2(7)	2(10)	4(8.2)	
Vasculopathy/vasculitis												
0	58(97)	43(98)	101(97.1)	1.000	26(96)	68(96)	94(95.9)	1	28(97)	19(95)	47(95.9)	1
1	2(3)	1(2)	3(2.9)		1(4)	3(4)	4(4.1)		1(3)	1(5)	2(4.1)	
Transverse Myelitis												
0	48(80)	40(91)	88(84.6)	0.172	18(67)	67(94)	85(86.7)	**0.001**	19(66)	18(90)	37(75.5)	0.089
1	12(20)	4(9)	16(15.4)		9(33)	4(6)	13(13.3)		10(34)	2(10)	12(24.5)	
AHNE												
0	51(85)	43(100)	94(91.3)	**0.010**	23(88)	65(92)	88(90.7)	0.698	25(86)	18(90)	43(87.8)	1
1	9(15)	0(0)	9(8.7)		3(12)	6(8)	9(9.3)		4(14)	2(10)	6(12.2)	
ADEM												
0	53(88)	41(93)	94(90.4)	0.512	23(85)	65(92)	88(89.8)	0.456	25(86)	17(85)	42(85.7)	1
1	7(12)	3(7)	10(9.6)		4(15)	6(8)	10(10.2)		4(14)	3(15)	7(14.3)	
CLOCC/MERS												
0	57(95)	38(86)	95(91.3)	0.163	27(100)	64(90)	91(92.9)	0.185	29(100)	18(90)	47(95.9)	0.162
1	3(5)	6(14)	9(8.7)		0(0)	7(10)	7(7.1)		0(0)	2(10)	2(4.1)	
** Others												
0	46(78)	33(77)	79(77.5)	1.000	24(89)	50(72)	74(77.1)	0.108	22(76)	18(90)	40(81.6)	0.277
1	13(22)	10(23)	23(22.5)		3(11)	19(28)	22(22.9)		7(24)	2(10)	9(18.4)	

CSF, cerebrospinal fluid; ADEM, acute disseminated encephalomyelitis; AHNE, acute hemorrhagic necrotizing encephalitis; CLOCC, cytotoxic lesion of the corpus callosum; MERS, mild encephalitis/ encephalopathy with reversible splenial lesion. * Variables take 1 if yes, take 0 if no. ** Other = PRES, posterior reversible encephalopathy syndrome; MOGAD, myelin oligodendrocytes glycoprotein antibody disease; delirium. ♦ Severity based on Infectious Diseases Society of America/American Thoracic Society guidelines.

**Table 5 brainsci-11-01354-t005:** Comparison of demographics, severity, fatality and CNS manifestation by serum markers.

	Serum d Dimer				CRP			
* Variables	Elevated	Normal	Total	Fisher Test	Elevated	Normal	Total	Fisher Test
	n (%)	n (%)	n (%)	*p*-Value	n (%)	n (%)	n (%)	*p*-Value
Age								
50+	19(66)	2(33)	21(60)	0.191	26(68)	2(17)	28(56)	**0.002**
<50	10(34)	4(67)	14(40)		12(32)	10(83)	22(44)	
Sex								
Male	23(79)	3(50)	26(74.3)	0.162	28(74)	7(58)	35(70)	0.471
Female	6(21)	3(50)	9(25.7)		10(26)	5(42)	15(30)	
♦ Severity								
Non-severe	9(31)	2(40)	11(32.4)	1	9(25)	8(80)	17(37)	**0.003**
Severe	20(69)	3(60)	23(67.6)		27(75)	2(20)	29(63)	
Fatality								
Non-fatal	24(80)	4(67)	28(77.8)	0.596	29(78)	9(82)	38(79.2)	1
Fatal	6(20)	2(33)	8(22.2)		8(22)	2(18)	10(20.8)	
Stroke								
0	27(90)	6(100)	33(91.7)	1	36(95)	12(100)	48(96)	1
1	3(10)	0(0)	3(8.3)		2(5)	0(0)	2(4)	
Encephalitis								
0	26(87)	4(67)	30(83.3)	0.256	28(74)	12(100)	40(80)	0.092
1	4(13)	2(33)	6(16.7)		10(26)	0(0)	10(20)	
Encephalopathy								
0	29(97)	5(100)	34(97.1)	1	34(89)	11(100)	45(91.8)	0.562
1	1(3)	0(0)	1(2.9)		4(11)	0(0)	4(8.2)	
Seizures								
0	30(100)	6(100)	36(100)	1	36(95)	12(100)	48(96)	1
1	0(0)	0(0)	0(0)		2(5)	0(0)	2(4)	
Vasculopathy/vasculitis								
0	29(97)	6(100)	35(97.2)	1	38(100)	9(75)		**0.011**
1	1(3)	0(0)	1(2.8)		0(0)	3(25)	3(6)	
Transverse Myelitis								
0	25(83)	4(67)	29(80.6)	0.573	33(87)	7(58)	40(80)	**0.046**
1	5(17)	2(33)	7(19.4)		5(13)	5(42)	10(20)	
AHNE								
0	22(73)	5(83)	27(75)	1	31(82)	11(92)	42(84)	0.661
1	8(27)	1(17)	9(25)		7(18)	1(8)	8(16)	
ADEM								
0	28(93)	6(100)	34(94.4)	1	36(95)	9(75)	45(90)	0.082
1	2(7)	0(0)	2(5.6)		2(5)	3(25)	5(10)	
CLOCC/MERS								
0	24(80)	6(100)	30(83.3)	0.561	30(79)	12(100)	42(84)	0.173
1	6(20)	0(0)	6(16.7)		8(21)	0(0)	8(16)	
** Others								
0	27(90)	4(67)	31(86.1)	0.186	32(84)	11(92)	43(86)	1
1	3(10)	2(33)	5(13.9)		6(16)	1(8)	7(14)	

Fisher exact test and Wilcoxon rank-sum test were used in the univariate data analysis for categorical and continuous variables, respectively, while logistic model was used in the multivariate data analysis. All statistical tests were two-sided and a *p*-value < 0.05 implies the statistical significance in this study. CSF, cerebrospinal fluid; ADEM, acute disseminated encephalomyelitis; AHNE, acute hemorrhagic necrotizing encephalitis; CLOCC, cytotoxic lesion of the corpus callosum; MERS, mild encephalitis/encephalopathy with reversible splenial lesion. * Variables take 1 if yes, take 0 if no. ** Other = PRES, posterior reversible encephalopathy syndrome; MOGAD, myelin oligodendrocytes glycoprotein antibody disease; delirium. ♦ Severity based on Infectious Diseases Society of America/American Thoracic Society guidelines.

**Table 6 brainsci-11-01354-t006:** Demographics and PNS manifestations in the 94 patients.

Variables	N (%)
**Age**	
>50	73 (77.7)
≤50	21 (22.3)
* **Sex**	
Male	59 (64.1)
Female	33 (35.9)
**PNS manifestation**	
GBS	89 (92.7)
Facial palsy	3 (3.1)
Polyneuritis cranialis	2 (2.1)
** **COVID-19 Severity**	
Non-severe	62 (67.4)
Severe	30 (32.6)
♦ **Outcome**	
Non-fatal	80 (88.9)
Fatal	10 (11.1)

* Gender unidentified in 2 cases, ** Severity not available in 2 cases, ♦ outcomes were not available in 4 cases; PNS, peripheral nervous system; GBS, Guillain–Barré syndrome.

**Table 7 brainsci-11-01354-t007:** Comparison of various PNS manifestation, demographics, severity and fatality by for CSF markers.

	CSF Protein				Elevated Cell Count				Lymphocytes			
	High	Low			Yes	No						
* Variables	>45	≤45	Total	Fisher Test	>5	≤5	Total	Fisher Test	>50%	≤50%	Total	Fisher Test
	n (%)	n (%)	n (%)	*p*-Value	n (%)	n (%)	n (%)	*p*-Value	n (%)	n (%)	n (%)	
Age												
50+	51(74)	15(88)	66(76.7)	0.338	4(67)	57(78)	61(77.2)	0.615	4(67)	15(79)	19(76)	0.606
<50	18(26)	2(12)	20(23.3)		2(33)	16(22)	18(22.8)		2(33)	4(21)	6(24)	
Sex												
Male	41(60)	11(69)	52(61.9)	0.582	2(40)	46(64)	48(62.3)	0.359	3(75)	14(74)	17(73.9)	1
Female	27(40)	5(31)	32(38.1)		3(60)	26(36)	29(37.7)		1(25)	5(26)	6(26.1)	
♦ Severity												
Non-severe	42(63)	13(76)	55(65.5)	0.395	4(67)	45(63)	49(63.6)	1	6(100)	11(65)	17(73.9)	0.144
Severe	25(37)	4(24)	29(34.5)		2(33)	26(37)	28(36.4)		0(0)	6(35)	6(26.1)	
Fatality												
Non-fatal	56(85)	17(100)	73(88)	0.114	5(83)	62(89)	67(88.2)	0.544	6(100)	14(82)	20(87)	0.539
Fatal	10(15)	0(0)	10(12)		1(17)	8(11)	9(11.8)		0(0)	3(18)	3(13)	
GBS												
0	3(4)	3(17)	6(6.8)	0.097	1(17)	5(7)	6(7.4)	0.38	3(43)	0(0)	3(11.5)	**0.013**
1	67(96)	15(83)	82(93.2)		5(83)	70(93)	75(92.6)		4(57)	19(100)	23(88.5)	
Facial palsy												
0	70(100)	16(89)	86(97.7)	**0.04**	6(100)	73(97)	79(97.5)	1	6(86)	19(100)	25(96.2)	0.269
1	0(0)	2(11)	2(2.3)		0(0)	2(3)	2(2.5)		1(14)	0(0)	1(3.8)	
Polyneuritis Cranialis												
0	68(97)	18(100)	86(97.7)	1	6(100)	73(97)	79(97.5)	1	6(86)	19(100)	25(96.2)	0.269
1	2(3)	0(0)	2(2.3)		0(0)	2(3)	2(2.5)		1(14)	0(0)	1(3.8)	

Fisher exact test and Wilcoxon rank-sum test were used in the univariate data analysis for categorical and continuous variables, respectively, while logistic model was used in the multivariate data analysis. All statistical tests were two-sided and a *p*-value < 0.05 implies the statistical significance in this study. CSF, cerebrospinal fluid; PNS, peripheral nervous system; GBS, Guillain–Barré syndrome. * Variables take 1 if yes, take 0 if no. ♦ Severity based on Infectious Diseases Society of America/American Thoracic Society guidelines.

**Table 8 brainsci-11-01354-t008:** Comparison of various for PNS manifestation, demographics, severity and fatality by serum markers.

	Serum d Dimer				CRP				Serum IL-6			
* Variables	Elevated	Normal	Total	Fisher Test	Elevated	Normal	Total	Fisher Test	Elevated	Normal	Total	Fisher Test
	n (%)	n (%)	n (%)	*p*-Value	n (%)	n (%)	n (%)	*p*-Value	n (%)	n (%)	n (%)	*p*-Value
Age												
50+	6(86)	2(100)	8(88.9)	1	5(100)	6(67)	11(78.6)	0.258	5(100)	0(0)	5(71.4)	0.048
<50	1(14)	0(0)	1(11.1)		0(0)	3(33)	3(21.4)		0(0)	2(100)	2(28.6)	
Sex												
Male	4(57)	1(50)	5(55.6)	1	2(40)	6(67)	8(57.1)	0.58	4(80)	2(100)	6(85.7)	1
Female	3(43)	1(50)	4(44.4)		3(60)	3(33)	6(42.9)		1(20)	0(0)	1(14.3)	
♦ Severity												
Non-severe	3(43)	2(100)	5(55.6)	0.444	3(60)	7(78)	10(71.4)	0.58	4(80)	2(100)	6(85.7)	1
Severe	4(57)	0(0)	4(44.4)		2(40)	2(22)	4(28.6)		1(20)	0(0)	1(14.3)	
Fatality												
Non-fatal	5(71)	2(100)	7(77.8)	1	5(100)	8(89)	13(92.9)	1	4(80)	2(100)	6(85.7)	1
Fatal	2(29)	0(0)	2(22.2)		0(0)	1(11)	1(7.1)		1(20)	0(0)	1(14.3)	
GBS												
0	0(0)	1(50)	1(11.1)	0.222	0(0)	2(22)	2(14.3)	0.505	0(0)	0(0)	0(0)	1
1	7(100)	1(50)	8(88.9)		5(100)	7(78)	12(85.7)		5(100)	2(100)	7(100)	
Facial palsy												
0	7(100)	1(50)	8(88.9)	0.222	5(100)	7(78)	12(85.7)	0.505	5(100)	2(100)	7(100)	1
1	0(0)	1(50)	1(11.1)		0(0)	2(22)	2(14.3)		0(0)	0(0)	0(0)	
Polyneuritis_Cranialis												
0	7(100)	2(100)	9(100)	1	5(100)	9(100)	14(100)	1	5(100)	2(100)	7(100)	1
1	0(0)	0(0)	0(0)		0(0)	0(0)	0(0)		0(0)	0(0)	0(0)	

Fisher exact test and Wilcoxon rank-sum test were used in the univariate data analysis for categorical and continuous variables, respectively, while logistic model was used in the multivariate data analysis. All statistical tests were two-sided and a *p*-value < 0.05 implies the statistical significance in this study. CSF, cerebrospinal fluid; PNS, peripheral nervous system; GBS, Guillain–Barré syndrome. * Variables take 1 if yes, take 0 if no. ♦ Severity based on Infectious Diseases Society of America/American Thoracic Society guidelines.

**Table 9 brainsci-11-01354-t009:** Comparison of various CSF markers, demographics, severity and fatality by PNS manifestation.

	GBS				Facial Palsy				Polyneuritis Cranialis			
* Variables	0	1	Total	Fisher Test	0	1	Total	Fisher Test	0	1	Total	Fisher Test
	n (%)	n (%)	n (%)	*p*-Value	n (%)	n (%)	n (%)	*p*-Value	n (%)	n (%)	n (%)	*p*-Value
CSF Protein												
>45	3(50)	67(82)	70(79.5)	0.097	70(81)	0(0)	70(79.5)	0.04	68(79)	2(100)	70(79.5)	1
≤45	3(50)	15(18)	18(20.5)		16(19)	2(100)	18(20.5)		18(21)	0(0)	18(20.5)	
CSF Cell Count												
>5	1(17)	5(7)	6(7.4)	0.38	6(8)	0(0)	6(7.4)	1	6(8)	0(0)	6(7.4)	1
≤5	5(83)	70(93)	75(92.6)		73(92)	2(100)	75(92.6)		73(92)	2(100)	75(92.6)	
CSF lymphocyte Percent												
>50	3(100)	4(17)	7(26.9)	0.013	6(24)	1(100)	7(26.9)	0.269	6(24)	1(100)	7(26.9)	0.269
≤50	0(0)	19(83)	19(73.1)		19(76)	0(0)	19(73.1)		19(76)	0(0)	19(73.1)	
♦ Severity												
Non-Severe	5(100)	57(66)	62(67.4)	0.169	60(67)	2(100)	62(67.4)	1	60(67)	2(100)	62(67.4)	1
Severe	0(0)	30(34)	30(32.6)		30(33)	0(0)	30(32.6)		30(33)	0(0)	30(32.6)	
Fatality												
Non-Fatal	6(100)	74(88)	80(88.9)	1	78(89)	2(100)	80(88.9)	1	78(89)	2(100)	80(88.9)	1
Fatal	0(0)	10(12)	10(11.1)		10(11)	0(0)	10(11.1)		10(11)	0(0)	10(11.1)	
Age												
≥50	1(20)	72(81)	73(77.7)	0.008	72(78)	1(50)	73(77.7)	0.399	73(79)	0(0)	73(77.7)	0.048
<50	4(80)	17(19)	21(22.3)		20(22)	1(50)	21(22.3)		19(21)	2(100)	21(22.3)	
Sex												
Male	3(75)	56(64)	59(64.1)	1	58(64)	1(50)	59(64.1)	1	58(64)	1(50)	59(64.1)	1
Female	1(25)	32(36)	33(35.9)		32(36)	1(50)	33(35.9)		32(36)	1(50)	33(35.9)	

Fisher exact test and Wilcoxon rank-sum test were used in the univariate data analysis for categorical and continuous. variables, respectively, while logistic model was used in the multivariate data analysis. All statistical tests were two-sided and a *p*-value < 0.05 implies the statistical significance in this study. CSF, cerebrospinal fluid; PNS, peripheral nervous system; GBS, Guillain–Barré syndrome. * Variables take 1 if yes, take 0 if no. ♦ Severity based on Infectious Diseases Society of America/American Thoracic Society guidelines.

## Data Availability

Data were extracted from the articles published in PUBMED, Google Scholar and Scopus. This will be provided on request.

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
