# Peer review of "Relevance of CSF, Serum and Neuroimaging Markers in CNS and PNS Manifestation in COVID-19: A Systematic Review of Case Report and Case Series"

_brainsci, 2021, doi:10.3390/brainsci11101354_

Round 1

Reviewer 1 Report

The effort taken by the authors to review the rising neurological manifestations post COVID-19 infection is highly appreciated. This review summarizes neurological manifestations in the CNS and PNS following COVID-19 infection. Following an exhaustive literature review,  the authors describe the occurrence of predominant CNS pathologies, namely transverse myelitis, AHNE (12, 10.4%), ADEM, CLOCC/MERS (10, 8.6%), and Vasculitis; and PNS pathologies like GBS following a COVID-19 infection. Data were evaluated using standard statistical measures, and the literature search criteria are well defined. This review fits with the scope of the journal. However, the manuscript lacks careful restructuring, consistency in data reporting, and flow of information, which is highly recommended to be revised before the manuscript is considered for publication. 

Overall, the manuscript is recommended for acceptance only after a thorough revision.

Here are specific comments and suggestions.

1. The authors are strongly encouraged to consolidate the results section under the broad heading “Results”, instead of “Results CNS” and “Results PNS”. Therefore, the findings under CNS category could be described under a sub-heading “CNS”. This will make more sense. The same goes for results described under “Results PNS”. 

Alternative: 

CNS manifestations of COVID-19

Followed by the sub-headings - Demographics, CSF Analysis, Severity, Fatality and Serum Markers

PNS manifestations of COVID-19

Followed by the sub-headings - Demographics, CSF Analysis, Severity, Fatality and Serum Markers

Neuroimaging observations of COVID-19 manifestation in CNS and PNS disorders

CNS Disorders

PNS Disorders

Discussion

2. For Serum Markers  (Sub-sections 3.1.5 and 3.2.3), please introduce the two markers (d-dimer and C-reactive protein) and their specificity in testing CNS/PNS disorder manifestation following COVID-19 infection. 

Note: The sequence in which this sub-section is described within the text is different for the CNS and the PNS disorders. Follow a consistent structure.

3. The data tables and figure should be formatted according to the manuscript guidelines. In the current available PDF version of the manuscript, the data tables are very hard to follow due to inappropriate formatting. Please correct these in the revision.

4. There are inconsistencies in the data reporting/writing in the result section. For example: 

In lines 132-140 the data is reported as (number, %); 141-153 the same type of information is presented as (n=no.of patients, %). For other paragraphs data is represented/written as (%).

The authors are suggested to adhere to a consistent reporting/writing style to make the review more readable.

Consider adopting the style in sections 3.1.3 and 3.1.4 for all the other sections. 

5. For a simplistic data representation, the authors are suggested to categorize types of CNS and PNS disorders as a pie-chart with percentages to provide a snapshot of the data and make the data presentation effective.

6. For the discussion - consider maintaining a flow of information. Each paragraph should convey a message without breaking the continuity of information, thereby communicating the information provided more effectively. 

The authors are recommended to revise the manuscript “critically” for grammar, sentence structure and formatting errors. 

Author Response

We would like to thank the reviewer for the time they invested in improving the quality of our manuscript. Please see attached point-by-point changes/corrections or explanations to the reviewers’ comments.

Reviewer#1

R#1.1: The authors are strongly encouraged to consolidate the results section under the broad heading “Results”, instead of “Results CNS” and “Results PNS”. Therefore, the findings under CNS category could be described under a sub-heading “CNS”. This will make more sense. The same goes for results described under “Results PNS”. 

Alternative: 

CNS manifestations of COVID-19

Followed by the sub-headings - Demographics, CSF Analysis, Severity, Fatality and Serum Markers

PNS manifestations of COVID-19

Followed by the sub-headings - Demographics, CSF Analysis, Severity, Fatality and Serum Markers

Neuroimaging observations of COVID-19 manifestation in CNS and PNS disorders

CNS Disorders

PNS Disorders

 A#1.1: Thank you for the comments. We have now categorized the results following the reviewer advice.

Please see the updated manuscript.

Discussion

R#1.2: For Serum Markers (Sub-sections 3.1.5 and 3.2.3), please introduce the two markers (d-dimer and C-reactive protein) and their specificity in testing CNS/PNS disorder manifestation following COVID-19 infection. 

Note: The sequence in which this sub-section is described within the text is different for the CNS and the PNS disorders. Follow a consistent structure.

 A#1.2: We have now included literature regarding the specificity of the serum markers.

Please see line # 288-296, & page # 11.

R#1.3: The data tables and figure should be formatted according to the manuscript guidelines. In the current available PDF version of the manuscript, the data tables are very hard to follow due to inappropriate formatting. Please correct these in the revision.

A#1.3: The tables are now aligned and edited to ensure clarity. Please let us know if this is appropriate.

R#1.4: There are inconsistencies in the data reporting/writing in the result section. For example: 

In lines 132-140 the data is reported as (number, %); 141-153 the same type of information is presented as (n=no.of patients, %). For other paragraphs data is represented/written as (%).

The authors are suggested to adhere to a consistent reporting/writing style to make the review more readable.

Consider adopting the style in sections 3.1.3 and 3.1.4 for all the other sections. 

A#1.4: We have now made changes following the style in the sections mentioned above.

Please see the updated manuscript.

R#1.5:  For a simplistic data representation, the authors are suggested to categorize types of CNS and PNS disorders as a pie-chart with percentages to provide a snapshot of the data and make the data presentation effective.

 A#1.5: Thank you for your recommendation. Pie charts for CNS and PNS disorders have been added. Please see the updated manuscript with pie chart.

R#1.6: For the discussion - consider maintaining a flow of information. Each paragraph should convey a message without breaking the continuity of information, thereby communicating the information provided more effectively. 

The authors are recommended to revise the manuscript “critically” for grammar, sentence structure and formatting errors. 

A#1.6: Thank you for the comments we now have made changes and also address the grammar changes as suggested.

Please see the updated manuscript.

Reviewer 2 Report

This is a very interesting review on an important topic. I have some comments and suggestions which may improve the quality of this review:

1. Introduction: Please define terminologies. The authors used manifestations. Please define and explain the differences to long-term, post-acute, and sub-acute. It is not clear, if the focus of this review is on acute COVID patients or not.

2. Results: Most tables and figures cannot be seen in the manuscript.

3. It is somewhat surprising that the authors only found MRI studies. What about PET?

4. Sex differences are not discussed at all. 

Author Response

We would like to thank the reviewer for the time they invested in improving the quality of our manuscript. Please see attached point-by-point changes/corrections or explanations to the reviewers’ comments.

Reviewer 2

This is a very interesting review on an important topic. I have some comments and suggestions which may improve the quality of this review:

R#2.1: Introduction: Please define terminologies. The authors used manifestations. Please define and explain the differences to long-term, post-acute, and sub-acute. It is not clear, if the focus of this review is on acute COVID patients or not.

A#2.1: Thank you for pointing this out. The appearance of neurological manifestations in COVID-19 patients range from 1 day to 8 weeks, therefore our focus has been on acute and sub-acute presentations, and not long-term presentation. The details of the timeline of onset of neurological symptoms is detailed in Supplementary tables 1 and 2. And line # 474-476, & page # 20.

R#2.2: Results: Most tables and figures cannot be seen in the manuscript.

A#2.2: We have now made the appropriate changes to ensure clarity.

R#2.3: It is somewhat surprising that the authors only found MRI studies. What about PET?

A#2.3:  While we reviewed findings of CT scan, PET scan and MRI scan, we only considered MRI scan findings in our quantitative analysis as data on other imaging modality were minimal.We have reported the relevant CT head findings for both CNS and PNS under the neuroimaging section. There seems to be only 1 study that used as PET neuroimaging, as reported by Delorme C et.al., a case series consisting of 4 patients with COVID-19 related encephalopathy with brain FDG-PET/CT pattern indicating frontal hypometabolism and cerebellar hypermetabolism. Please see line # 448-451, & page # 19.

R#2.4: Sex differences are not discussed at all

A#2.4: Thank you for this comment. Even though we did not explicitly state in the results section about sex differences, we have included differences in sex with various presentations and lab studies in the respective tables (Table number 2, 4 & 5).

Please see line # 177-184, & page # 5.

Round 2

Reviewer 2 Report

The authors responded to my comments very well. Thank you.